# Novel Biomarkers of Diabetic Kidney Disease

**DOI:** 10.3390/biom13040633

**Published:** 2023-03-31

**Authors:** Jorge Rico-Fontalvo, Gustavo Aroca-Martínez, Rodrigo Daza-Arnedo, José Cabrales, Tomás Rodríguez-Yanez, María Cardona-Blanco, Juan Montejo-Hernández, Dairo Rodelo Barrios, Jhonny Patiño-Patiño, Elber Osorio Rodríguez

**Affiliations:** 1Nephrologist, Colombian Association of Nephrology, Bogota 110221, Colombia; 2Faculty of Medicine, Universidad Simón Bolívar, Barranquilla 080002, Colombia; 3Nephrology Fellow, Stanford University School of Medicine, Palo Alto, CA 94305, USA; 4Internal Medicine, Universidad de Cartagena, Cartagena 130001, Colombia; 5Graduate Student in Critical Medicine and Intensive Care, Simon Bolivar University, Barranquilla 080002, Colombia; 6Independent Researcher, Barranquilla 080002, Colombia; 7Group of Intensive Care and Comprehensive Care (GRIMICI), Barranquilla 080002, Colombia

**Keywords:** biomarkers, diabetic kidney disease, pathogenesis, diabetes mellitus

## Abstract

Diabetic kidney disease (DKD) is a highly prevalent condition worldwide. It represents one of the most common complications arising from diabetes mellitus (DM) and is the leading cause of end-stage kidney disease (ESKD). Its development involves three fundamental components: the hemodynamic, metabolic, and inflammatory axes. Clinically, persistent albuminuria in association with a progressive decline in glomerular filtration rate (GFR) defines this disease. However, as these alterations are not specific to DKD, there is a need to discuss novel biomarkers arising from its pathogenesis which may aid in the diagnosis, follow-up, therapeutic response, and prognosis of the disease.

## 1. Introduction

Diabetic kidney disease (DKD) is a highly prevalent condition worldwide. It represents one of the most frequent complications of diabetes mellitus (DM) and is the primary cause of end-stage kidney disease (ESKD). Its pathogenesis involves three fundamental components: the hemodynamic, metabolic, and inflammatory axes [1]. Persistent albuminuria, accompanied by a progressive decline in glomerular filtration rate (GFR) [2], characterizes DKD clinically. However, these alterations are not specific to DKD, highlighting the need to identify novel biomarkers that arise from the pathogenesis of this disease to aid in its diagnosis, follow-up, therapeutic response, and prognosis.

While there are several reviews that have examined the inflammatory component of DKD, few have provided a comprehensive review that is specific to DKD. Therefore, this paper aims to provide a unique and comprehensive review of the inflammatory component of DKD, highlighting the current understanding of the pathophysiology of inflammation in the development and progression of DKD.

As diabetes mellitus is an inflammatory disease that extends beyond the traditional hemodynamic and metabolic axes [3], the early detection of inflammatory biomarkers is essential in reducing complications related to the disease. Early detection, coupled with the optimization of currently available therapeutic options, can delay the progression of DKD towards terminal stages that require renal replacement therapy and cause death [1,4]. Therefore, this review aims to analyze the molecular aspects of the disease and the clinical utility and role of new biomarkers in the management of DKD.

## 2. Epidemiology and Perspectives in Diabetic Kidney Disease

Diabetes mellitus (DM) is one of the most common chronic, non-communicable diseases, making it one of the fastest-growing global health emergencies in recent decades. According to recent data from the International Diabetes Federation (IDF), the global prevalence of DM in adults aged 20 to 79 years was 537 million in 2021, representing 10.5% of the population. This number is expected to rise to 643 million by 2030 [5]. However, there is often a delay of 4 to 7 years between the onset of DM and its diagnosis, and even more time passes before any clinical damage becomes apparent. Therefore, the development of earlier diagnostic strategies is of great interest.

Diabetic kidney disease (DKD) affects approximately 30–40% of patients with type 1 or type 2 diabetes mellitus (DM) [6]. The pathogenesis and progression of DKD involve three fundamental axes: the hemodynamic, metabolic, and inflammatory axes. Of these, the inflammatory axis is gaining increasing importance and evidence as a possible therapeutic target [1].

Traditionally, DKD has been considered a non-inflammatory glomerular disease, with its pathogenesis attributed to hemodynamic and metabolic alterations. However, recent discoveries and advances in understanding the immune response have challenged this view, identifying inflammation as a central phenomenon in the development and progression of DKD [1]. This updated understanding of the disease has led to advances in its treatment and has opened up the possibility of considering new molecules identified in the course of the disease as therapeutic targets and markers for diagnosis, monitoring, and prognostication. Figure 1 reviews the various components of the pathophysiology of DKD.

## 3. Limitations of Conventional Biomarkers

The diagnosis of DKD relies on the detection of albuminuria and a decrease in the estimated glomerular filtration rate (eGFR), which is measured by serum creatinine levels [7]. The decline in the eGFR is a consequence of renal function loss and significant destruction of the glomeruli [7]. Among the markers used to assess DKD, albuminuria is typically the most robust tool for predicting prognosis and evaluating treatment efficacy [5]. Urinary albumin excretion (UAE) is utilized for risk classification and reflects the severity of albuminuria [8].

There is a clear correlation between the extent of structural damage and renal function, particularly at moderate levels of eGFR reduction during DKD [9]. However, this association is less apparent in the initial stages of the disease when albuminuria is low or eGFR reduction is minimal [9]. This highlights the need to identify biomarkers that enable earlier evaluation of renal structural damage and the identification of populations at a high risk of progression [9].

In DKD follow-up, the main predictors of disease progression are variations in the current and previous eGFRs [10]. Therefore, the GFR remains the primary clinical biomarker for evaluating prognosis in DKD and is frequently used in clinical practice and trials [10]. The CKD-EPI and MDRD equations, which use creatinine as a marker to estimate the GFR, are commonly used for GFR estimation [10]. The differences in the diagnostic accuracy of these two equations are minimal. However, it is important to note that these equations may either under- or overestimate the true GFR in DKD due to compensatory changes in the remaining nephrons. An important limitation of the currently available diagnostic and follow-up methods is their reduced utility in patients with a normoalbuminuric phenotype, which is increasingly prevalent and lacks targeted therapy [11].

Given the limitations of currently available diagnostic and follow-up methods in patients with normoalbuminuric phenotype, there is a need to develop and identify new biomarkers for DKD. We propose to group these biomarkers based on their utility in diagnosing, monitoring progression, evaluating therapeutic response, and predicting prognosis in patients with DKD, as shown in Table 1**.** We will review information on novel biomarkers and their potential clinical utility in these scenarios.

## 4. Novel Biomarkers in Diabetic Kidney Disease

Undoubtedly, early diagnosis is crucial for risk management to improve prognosis and slow the progression of kidney disease associated with DM [1,23,24]. However, conventional diagnostic tests have profound limitations, as previously described, which often result in a late diagnosis. Therefore, the continuous search for novel biomarkers in urine and serum, based on proteomics and metabolomics, has gained relevance in recent years. This is due to the improved understanding of the pathophysiology and mechanisms associated with the development of DKD [23,25].

Due to advances in understanding the molecular mechanisms behind acute and chronic kidney injury, numerous biomarkers have been proposed. While these have mainly been identified in acute kidney injury (AKI), they could also have significance in the evaluation of patients with diabetic kidney disease (DKD). Such biomarkers could increase the precision with which the presence and progression of DKD is evaluated, an area in which conventional biomarkers, such as serum creatinine, used to estimate GFR and albuminuria have limitations [26,27,28]. Despite the availability of novel biomarkers with a high diagnostic and discriminative capacity, good sensitivity and specificity, and the capability of detecting minimal changes in renal structure and function, the evidence of their usefulness remains variable and controversial [26,29].

Moreover, it is important to note that as per the AKI consensus, biomarkers do not replace conventional tests and adequate clinical evaluation. Rather, they serve as complementary tests that allow for the earlier identification and individualization of populations that could potentially benefit from cardiovascular risk prevention and management interventions [30]. In the following sections, we describe the biomarkers proposed in different scenarios in DKD.

### 4.1. Diagnostic Biomarkers

The conventional diagnostic tests for DKD, as previously mentioned, include albuminuria and the eGFR, which is estimated by creatinine [8]. However, several clinical factors have been identified that can predict the decline in glomerular filtration rate and progression of DKD [10]. These factors include age, duration of diabetes, levels of glycosylated hemoglobin (HbA1c), systolic blood pressure (SBP), albuminuria, previous eGFR, and the presence of other microvascular complications such as diabetic retinopathy [8,10]. Despite the existence of these factors, robust predictive equations are not available to predict the development of DKD, highlighting the need for improved risk management beyond the knowledge derived from epidemiologically based studies on the risk of eGFR decline associated with these factors [31,32]. Furthermore, given the limitations in the use of creatinine and albuminuria and the existence of the normoalbuminuric phenotype of DKD, other diagnostic tests are necessary to identify the presence of DKD.

#### 4.1.1. Serum Cystatin C (CysC)

Cystatin C (CysC) is a low-molecular-weight protein composed of 122 amino acids and is part of the cysteine protein inhibitor family. It is encoded by the CST3 housekeeping gene located on chromosome 20 (20pl1. 21) [33]. Cysteine cathepsins play a role in various physiological processes, including protein turnover, pro-protein processing, bone remodeling, antigen presentation, and apoptosis [34]. CysC, which is the most abundant and potent member of the endogenous inhibitors that control the activities of these enzymes both inside and outside cells, participates in numerous pathological processes, such as cardiovascular disease and inflammation [35].

CysC is synthesized and released into the plasma at a constant rate by all nucleated cells in the organism. Due to its small size and positive charge, it is freely filtered at the glomerular level and then completely reabsorbed and degraded but not secreted by the renal tubules [36,37]. Therefore, serum CysC can be used as a biomarker for the early diagnosis of AKI and may reflect early changes in renal function and reduction in the eGFR. In scenarios such as the first 6 hours in the postoperative period of cardiovascular, it has a sensitivity and specificity estimated at 71% and 53%, respectively [13]. CysC enables an earlier diagnosis of AKI than creatinine variations, but its use is limited when compared with other biomarkers such as neutrophil gelatinase-associated lipocalin (NGAL) [38,39].

In a prospective observational study involving 237 patients with type 2 diabetes mellitus, both serum and urinary levels of CysC were evaluated over a follow-up period of 29 months. The study showed that CysC levels were associated with both a decrease in the eGFR and the progression of DKD [40]. Further analysis of this study suggested that CysC could be an earlier marker of a reduced filtration rate compared to creatinine. Another study by Jeon et al. also demonstrated that CysC levels increased with increasing stages of CKD from I to III and from normoalbuminuria to microalbuminuria, with a positive correlation with the albumin-to-creatinine ratio [41].

#### 4.1.2. Neutrophil Gelatinase-Associated Lipocalin (NGAL)

NGAL, also known as neutrophil gelatinase-associated lipocalin, is a 25 kDa protein belonging to the lipocalin family [42]. It is produced by both neutrophils and injured epithelial cells of the nephron, leading to its specific release into the blood and urine in response to nephron injury [42]. NGAL is filtered from plasma through the glomerular filtrate and then reabsorbed by endocytosis via the megalin system in the proximal tubule.

In a cross-sectional study of 94 diabetic patients and 45 non-diabetic control subjects, the levels of serum and urinary damage markers were measured, revealing that NGAL levels were 1.5 times higher in the diabetic patients than in the healthy subjects. This study also found that markers of glomerular and tubular damage were associated with the presence of albuminuria independent of eGFR, suggesting that albuminuria could result from both glomerular and tubulointerstitial damage [43]. NGAL has also been proposed as a potential biomarker for identifying and detecting early-stage DKD [14].

Tubular injury may occur before glomerular injury in people with diabetes, and NGAL can be a useful biomarker for early detection of diabetic nephropathy (DN). NGAL can detect incipient nephropathy changes earlier than proteinuria. In a study of 144 patients with type 2 diabetes mellitus, both serum and urinary measurements of NGAL were performed, and both were able to predict the appearance of albuminuria, allowing for the early detection of DN [14]. Additionally, Carvalho et al. found that uKIM-1 and urinary NGAL levels were increased in type 2 diabetes mellitus patients with normal or mildly increased albuminuria, suggesting that both tubular and glomerular injuries may be occurring even at the earliest stages of DKD [44].

A recent meta-analysis that included 19 studies found that serum NGAL had a pooled sensitivity of 0.79 (95% confidence interval [CI] 0.60–0.91) and a specificity of 0.87 (0.75–0.93). Urine NGAL had a pooled sensitivity of 0.85 (0.74–0.91) and a specificity of 0.74 (0.57–0.86) [45]. The pooled sensitivity and specificity for kidney disease in normoalbuminuric patients with diabetes were 0.90 (0.82–0.95) and 0.97 (0.90–0.99) for serum NGAL, respectively, and 0.94 (0.87–0.98) and 0.90 (0.81–0.96) for urine NGAL, respectively [45]. These results indicate that NGAL can be useful for classifying DKD and can provide an added diagnostic value in the group of patients with normoalbuminuric kidney disease [45].

#### 4.1.3. Plasma KIM-1 (KIM-1)

Injury molecule 1 (KIM-1) is a type I transmembrane glycoprotein that is expressed in the apical membrane of the proximal renal tubular cells; its serum levels tend to increase in patients with tubular injury [15]. In a cohort study of patients with type 1 diabetes mellitus and proteinuria, baseline serum KIM-1 levels were a strong predictor of eGFR loss and ESKD during the 5 to 15 years of follow-up after adjusting their values for baseline urinary albumin-to-creatinine ratio levels, eGFR, and Hb1Ac [46]. Additionally, in a cohort study that included 462 patients, of whom 259 were normoalbuminuric and 203 of had microalbuminuria, plasma KIM-1 levels predicted an early reduction in eGFR and the progression of kidney disease independent of other variables, such as systolic blood pressure (BP), HbA1c, AER, eGFR, and TNFR1 [12]. On the contrary, data on urinary KIM-1 in DKD were disappointing and limited thus far [15].

#### 4.1.4. Other Diagnostic Biomarkers in DKD

There are several biomarkers with probable utility in the context of diabetic kidney disease (DKD), including fibroblast growth factors 21 and 23 (FGF21, FGF23) and pigment epithelium-derived factor (PEDF), which are associated with inflammatory pathways and fibrosis [10]. Additionally, markers of endothelial dysfunction, such as the mid-regional fragment of pro-adrenomedullin (MR-proADM), and cardiac injury markers, such as N-terminal pro-B-type natriuretic peptide (NT-proBNP), may correlate with a decline in the eGFR [10]. Copeptin, a biomarker derived from arginine vasopressin, has been associated with the progression of DKD and is an independent marker of eGFR reduction and even progression to end-stage renal disease (ESRD) [10]. While the list of biomarkers with probable utility in DKD continues to grow, their validation in clinical scenarios has been limited.

### 4.2. Biomarkers of Therapeutic Response

In the treatment and risk management of patients with DKD, it is crucial to search for biomarkers that can predict an individual’s response to proposed treatments as well as evaluate their risk prediction. However, many studies that evaluate and predict disease progression do not provide information regarding the participants’ response to the evaluated drugs, nor do they identify patients at an increased risk of secondary or adverse events from the therapeutic intervention studied [6]. Additionally, individual patients can show a great variability in their clinical response to drugs or therapeutic interventions [6]. Therefore, having biomarkers of therapeutic response can save time and reduce costs when prescribing a drug [6].

Conversely, identifying patients with a higher probability of responding to therapeutic and reno-protective strategies can modify the course of the disease and optimize the impact of the therapeutic resources available. The use of biomarkers to stratify patients according to their response to treatment is an encouraging idea [17]. Measuring these biomarkers before exposing the patient to the intervention and defining whether therapy is indicated based on the biomarker level is an approach that has gained ground in the management of DKD. Short-term changes in biomarkers are then used to predict the long-term efficacy of the drug [17].

Markers of clinical response commonly used in different studies include the reduction of albuminuria, blood pressure, glucose, or cholesterol levels [47]. However, these markers alone may not be sufficient to define populations with potential benefits from drug use before initiation or an increased risk of side effects [47]. Surrogates of therapeutic response have shown limitations in study designs evaluating the real effect of novel drugs, such as SGLT2 inhibitors, or previous interventions, such as aldosterone receptor antagonists, in which the polymorphism of the angiotensin-converting enzyme II can predict populations with a greater or lesser response to pharmacological intervention [47]. Therefore, defining better markers of therapeutic response presents a clinical challenge.

For instance, SGLT2 inhibitors have been shown to be innovative drugs with favorable clinical outcomes in patients with DKD [16,48]. Conventional markers such as glycosylated hemoglobin may not fully reflect the response to treatment, and clinical markers such as the reduction in blood pressure and albuminuria behavior may even reflect the long-term therapeutic response [17,49]. Thus, it is essential to find more precise markers that can better predict the therapeutic response to SGLT2 inhibitors and other treatments for DKD.

In the CANTATA-SU study comparing canagliflozin against glimepiride, a subsequent analysis of patient samples found that the use of canagliflozin led to significant reductions in the plasma levels of TNF receptor 1 (TNFR1; 9.2%; *p* < 0.001), IL-6 (26.6%; *p* = 0.010), matrix metalloproteinase 7 (MMP7; 24.9%; *p* = 0.011), and fibronectin 1 (FN1; 14.9%; *p* = 0.055) during the two-year follow-up period [50,51]. These markers of inflammation and extracellular matrix alteration may serve as indicators of therapeutic response to iSGLT-2 treatment. In addition, a sub-analysis of the Canagliflozin Cardiovascular Assessment Study (CANVAS) showed that canagliflozin use decreased KIM-1 and moderately reduced TNFR-1 and TNFR-2 compared to a placebo in people with type 2 diabetes [52]. Dapagliflozin, another SGLT2i, has also been studied for its effects on the urine metabolome and was found to increase urinary concentrations of branched-chain amino acids, lactate, and ketone bodies, which contribute to reno-protective effects. Moreover, dapagliflozin use resulted in a significant clinical reduction of albuminuria in patients with DKD [53].

In addition to iSGLT-2 inhibitors, other pharmacological interventions are being studied to identify biomarkers of therapeutic response, including pentoxifylline. In the PREDIAN study, it was observed that adding pentoxifylline to RAS blockade therapy led to a reduction in the urinary excretion of TNF-α, which was inversely related to the eGFR in patients with stage 3 and 4 chronic kidney disease [54]. This suggests that TNF-α could potentially serve as a biomarker for monitoring the therapeutic response to pentoxifylline treatment in patients with CKD.

Proteomics has the potential to facilitate personalized medicine by identifying patients who are more likely to respond to treatment and are at greater risk of adverse or secondary events [6]. Evaluating an individual’s proteome can aid in determining patients with a favorable or poor baseline response to therapeutic intervention [55]. Additionally, monitoring the proteome dynamically during treatment could identify patients who experience additional benefits or favorable clinical results over time that could be adjusted based on their response to treatment [6].

Although proteomics holds great potential in identifying new biomarkers of therapeutic response, studies in this area are often limited by their high cost and small sample sizes. Methodological biases, such as the type of study design used or the lack of adjustment for confounding factors such as baseline eGFR, further complicate the analysis of proteomic data [15]. Currently, CKD-273 is the proteomic biomarker with the strongest evidence for its use in managing DKD [15]. This biomarker has been suggested to identify patient subgroups that are more likely to respond to therapeutic interventions, such as ARB II or spironolactone, in studies such as DIRECT-2, PRIORITY, and the MAR-LINA-T2D trial. However, despite this promising potential, these studies have not yet demonstrated clear benefits in reducing the progression of kidney disease or albuminuria [56,57,58]. More research is needed to fully evaluate the clinical utility of proteomic biomarkers in the management of DKD.

The study of pathophysiological mechanisms during DKD, particularly inflammation, has identified potential therapeutic targets and prognostic markers that offer opportunities for increasing treatment options for these patients [1]. Of particular interest is the activation of the neutrophil mechanisms involved in the innate immune response, which leads to DNA de-condensation and histone citrullination by PAD-4, a histone deaminase [18]. This results in the secretion of NETs, which contain DNA, histones, and neutrophil proteases such as neutrophil elastase (NE) and myeloperoxidase (MPO) [18]. NETs have been proposed as mechanisms that mediate the development of cardiovascular diseases and DKD through inflammation that is not mediated by an infection response. Recent research has suggested that inhibiting NETs could be a novel therapeutic target and a biomarker for follow-up and prognosis in DKD patients [18].

The identification and study of biomarkers for therapeutic response is an ongoing process, utilizing proteomic and metabolomic analysis to identify clinically relevant molecules.

### 4.3. Monitoring and Prognostic Biomarkers

We can begin by discussing biomarkers of tubular damage, such as KIM-1, NGAL, α-1-microglobulin, NAG, cystatin C, and L-FABP [2]. KIM-1 is one of the most-studied markers, with research including a nested case–control study (*n* = 190 cases of incident DKD and 190 matched controls) and a prospective cohort study (*n* = 1156) that used banked baseline plasma samples from participants in randomized controlled trials of early (ACCORD) and advanced (VA NEPHRON-D) DKD [59]. The results of these studies indicate that KIM-1 is a strong predictor of decline in glomerular filtration rate in both early and advanced DKD [59]. In addition, in the Chronic Renal Insufficiency Cohort (CRIC) Study, which involved 894 patients, high plasma levels of KIM-1, TNFR-1, TNFR-2, MCP-1, suPAR, and YKL-40 were associated with an increased risk of DKD progression, with TNFR-1 being discussed in further detail in another section [60]. However, there is still a need for further research on biomarkers of therapeutic response, including proteomic and metabolomic responses, to identify molecules of clinical interest.

Liver-type fatty acid-binding protein (L-FABP) is another marker of tubular damage and DKD progression. Elevated levels of L-FABP in urine have been shown to correlate with the progression of DKD [2]. In a study that included 1549 patients with type 1 diabetes, 334 had microalbuminuria, 363 had macroalbuminuria, and the remaining patients had an albumin excretion rate (AER) within normal reference ranges [19]. The study found that elevated levels of L-FABP were an independent factor in the progression of DKD, regardless of the clinical stage of the patients [19]. In contrast, other markers of tubular damage, such as NGAL or cystatin C, have shown controversial results in predicting DKD progression [2].

It has been determined that inflammation markers such as tumor necrosis factor alpha (TNF-α) and IL-1β could predict the progression of diabetic nephropathy [61,62]. TNF-α binds to TNF-α type 1 (TNFR1) and type 2 (TNFR2) receptors. The latter can be identified in the circulation in soluble form [2]. On the other hand, TNF-α exerts cytotoxic effects on glomerular and mesangial cells and is therefore a determinant of the development of renal injury [1,63]. Its elevated levels have been identified in animal and human models of diabetic nephropathy at both urinary and serum levels [2,20]. Great interest has also been aroused in the presence of high levels of soluble forms of TNF-α receptors in the systemic circulation, as discussed below [2].

Tumor serum necrosis receptor factors 1 (TNFR1) and 2 (TNFR2) are proinflammatory markers that are increasingly used in monitoring and predicting the progression of DKD. Elevated levels of TNFR1 and TNFR2 are associated with worsening albuminuria, a reduced GFR, the onset of end-stage renal disease, the requirement of renal replacement therapy, and death [21,64,65,66]. Furthermore, serial histopathological studies have demonstrated that the presence of elevated levels of TNFR1 and TNFR2 are strongly related to the development of end-stage renal disease in patients with type 2 DM and are associated with early changes in the clinical course of DKD [21]. Among these early changes, a significant inverse proportional correlation was found with the percentage of endothelial cell fenestration and the total filtration surface of the glomerulus. Additionally, a significant positive correlation was made with the mesangial fractional volume, glomerular basement membrane width, podocyte process width, and the percentage of global glomerular sclerosis [21]. These findings support the notion that the inflammatory activity of these molecules is associated with early changes in DKD.

The Diabetes Control and Complications Trial (DCCT) was a population study that analyzed 1237 patients who were free of albuminuria and cardiovascular disease at the beginning of the follow-up period [67]. A sub-analysis of this study determined that high levels of inflammatory markers, especially E-selectin (sE-selectin) and soluble tumor necrosis factor receptors (TNFR)-1 and -2, are important determinants of the occurrence of microalbuminuria in patients with type 1 diabetes mellitus [67]. In a complementary observational cohort study that included 349 patients with type 1 diabetes mellitus, high circulating levels of TNFR2 were identified as the main determinants of the decline in the eGFR in patients with type 1 diabetes mellitus and proteinuria. These effects are accentuated by poor metabolic control associated with elevated levels of glycosylated hemoglobin [68].

In addition, individual plasma markers of inflammation and fibrosis, such as TNFR1, TNFR2, and YKL-40, as well as tubular damage markers such as KIM-1 have been associated with an increased risk of end-stage renal disease requiring renal replacement therapy in adults with DM and GFR < 60 mL/min/1.73 m^2^ [69]. In individual biomarker models adjusted for eGFR, UACR, and established risk factors, hazard ratios for incident kidney failure with replacement therapy (KFRT) per 2-fold higher biomarker concentrations were 1.52 (95% CI, 1.25–1.84) for plasma KIM-1, 1.54 (95% CI, 1.08–2.21) for TNFR1, 1.91 (95% CI, 1.16–3.14) for TNFR2, and 1.39 (95% CI, 1.05–1.84) for YKL-40 [69]. These biomarkers have shown promising potential for identifying populations at a high risk for poor disease progression and requiring renal replacement therapy.

The increasing knowledge of the pathophysiological mechanisms involved in the clinical course of DKD has allowed for the identification of other pathological pathways beyond metabolic control and hemodynamic alterations. In addition to biomarkers associated with inflammation, we will comment on those derived from oxidative stress [1]. One such biomarker is 8-Hydroxy-2′-deoxyguanosine (8-OHdG), a product of oxidative DNA damage that can be found at the plasma level and is excreted in urine after DNA repair by nuclease activity [2,70]. This suggests that 8-OHdG could be a biomarker of oxidative DNA damage in diabetic patients, a group which has demonstrated higher levels of this product when compared to healthy populations [71].

It is important to note that while some studies have shown the potential utility of 8-OHdG as a prognostic marker of DKD, other studies have reported limitations in its use [22]. For instance, one study found that monitoring the kinetics of 8-OHdG at the urinary level may be challenging in establishing a prognosis [72]. Therefore, further research is needed to better understand the role of oxidative stress biomarkers, including 8-OHdG, in predicting the adverse outcomes and progression of DKD. Nonetheless, these findings highlight the importance of considering other pathological pathways beyond metabolic control and hemodynamic alterations in the pathogenesis of DKD.

The identification of biomarkers related to tubular damage, inflammation, fibrosis, and oxidative stress shows promise in identifying high-risk populations for DKD and developing effective intervention and prevention strategies. However, further studies are necessary to establish the clinical relevance of these biomarkers and to monitor the role of novel molecules developed for diabetes mellitus treatment. Figure 2 provides an overview of the potential biomarkers for DKD.

## 5. Limitations in the Use of New Biomarkers

The history of medicine has brought us to a point at which personalized intervention strategies are being sought with the aim of identifying molecules that can help classify subpopulations and optimize therapeutic interventions. However, in the case of biomarkers for DKD, much of the available information is based on animal, cellular, and in vitro studies, and the results of large population studies are controversial [10].

Another crucial factor to consider is the cost-effectiveness of using these biomarkers, which is beyond the scope of this publication. The limited availability of these markers worldwide restricts their widespread use. Moreover, it is crucial to consider the potential variability resulting from diagnostic test studies and the performance of each test, which may alter the interpretation of the results [73].

It is important to note that the diagnostic performance of these biomarkers is not limited to DKD and can be influenced by other factors, such as infectious processes, which can affect molecules such as IL-18, NGAL, and calprotectin without necessarily indicating the presence of acute kidney injury [74]. Therefore, the impact of comorbidities on the behavior of these biomarkers should be taken into consideration, and the proposed cut-off points for diagnosis, follow-up, and prognosis may need to be re-evaluated [75].

Moreover, recent studies have revealed that DM can affect the performance of biomarkers in predicting the development of kidney disease, adding to the challenge of identifying suitable biomarkers for diagnosis, therapeutic response, follow-up, and prognosis [76]. However, the application of these biomarkers is limited by various factors such as the availability of test platforms, cost, variability in testing techniques and results, and the lack of approval from national and international regulatory bodies [77].

## 6. Conclusions

In recent years, there has been an emergence of new treatment options for DKD, and research is ongoing to identify potential therapeutic targets for prevention, follow-up, and prognosis. However, the implementation of these biomarkers in clinical practice is still a work in progress, as further studies are needed to confirm their utility. Nonetheless, the development of novel molecules and treatment strategies is promising, and it is hoped that in the future, clinicians will have access to a wider range of effective and personalized interventions for patients with DKD.

## Figures and Tables

**Figure 1 biomolecules-13-00633-f001:**
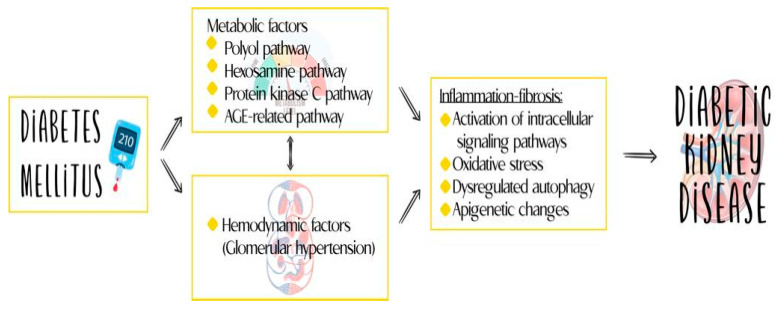
Inflammation plays an important role in the pathogenesis of DKD.

**Figure 2 biomolecules-13-00633-f002:**
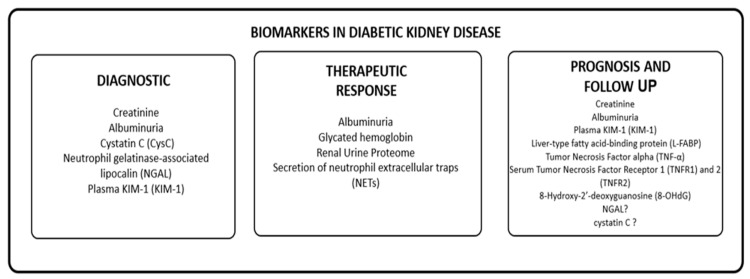
Biomarkers of DKD.

**Table 1 biomolecules-13-00633-t001:** Biomarkers in diabetic kidney disease.

Author	Year	Biomarker	Diagnostic	Therapeutic Response	Monitoring and Forecasting	Comment
Pena, M.J. [5]	2016	Renal urinary proteome	0	1	0	Could identify who will have the best clinical outcome
Fontalvo, J.E.R. [8]	2021	Albuminuria	1	1 [12]	1	Urinary measurement
Colhoun, H.M. [10]	2018	Creatinine	1	0	1	Plasma measurement
Haase-Fielitz, A. [13]	2009	Serum Cystatin C (CysC)	1	0	1 *	Marker ↑ before creatinine variation
Kaul, A. and Behera, M.R. [14]	2018	NGAL	1	0	1 *	Serum and urinary measurement
Barutta, F. [15]	2021	Plasma KIM-1	1	0	1 [16]	Serum measurement
Tye, S.C. [17]	2021	Glycosylated hemoglobin	0	1 *	0	Serum measurement
Gupta, A. [18]	2022	NETs	0	1	0	Inhibition may be a therapeutic target
Panduru, N.M. [19]	2013	L-FABP	0	0	1	Urinary measurement
Kalantarinia, K. [20]	2014	TNF-a	0	0	1	Serum and urinary measurement
Pavkov, M.E. [21]	2015	TNFR 1 -2	0	0	1	Plasma measurement
Sanchez, M. [22]	2018	8-OHdG	0	0	1	Plasma measurement

Diagnosis—0 = no, 1 = yes; Therapeutic Response—0 = no, 1 = yes; Follow-up and Prognosis—0 = no, 1 = yes; NGAL—Neutrophil gelatinase-associated lipocalin; NETs—secretion of neutrophil extracellular traps; L-FABP—liver-type fatty acid-binding protein TNF-α—tumor necrosis factor alpha; TNFR 1-2—serum tumor necrosis factor receptors 1 and 2; 8-OHdG—8-Hydroxy-2′-deoxyguanosine; *—controversial results.

## Data Availability

Not applicable.

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
