# Peer review of "Novel Biomarkers of Diabetic Kidney Disease"

_biomolecules, 2023, doi:10.3390/biom13040633_

Round 1

Reviewer 1 Report

To the Editor of Biomolecules

It was with interest that I read Rico-Fontalvo’s et al. manuscript on “Novel Biomarkers of diabetic kidney disease”, a very important topic due to the huge clinical and economic costs associated with ESKF and diabetes in the diabetic kidney disease.

It is fundamental to diagnose as soon as possible the initial damage of diabetes also in the kidney, so as to be able to implement all the available therapeutic resources today.

COMMENTS:

Abstract: (lines 18 and 20) please do not insert references in the abstract.

Introduction: (line 50) ..there is usually a delay of 4 to 7 years between the onset of the disease (diabetes) and its diagnosis”: ..and an even longer time before seeing the clinical kidney damage

3. Limitations of coventional biomarkers. (pag 2, line 86). I agree that the differences in the diagnostic accuracy of different equations for eGFR are minimal. However, the authors have to take into account that the risk of DKD development or progression may be under or overesti­mated depending on which eGFR equation is used for risk stratification because eGFR is related to number of functioning nephrons. So a reduction in the number of nephrons due to the progression of DKD might be compensated by an increase in the single nephron GFRs of surviving nephrons and these changes might not be accurately reflected by the eGFR.

4.1.1 Serum cystatin C: (page 3, lines 135-143) The introduction about CysC in kidney disease and AKI is too long, while the discussion on its role in DKD is confined only in the last 6 lines..

4.1.2 NGAL: if the space available allowed it, it would be interesting to include the work of Carvalho (de Carvalho JA, Tatsch E, Hausen BS, et al. Urinary kidney injury molecule-1 and neutrophil gelatinase-associated lipocalin as in­dicators of tubular damage in normoalbuminuric patients with type 2 diabetes. Clin Biochem 2016;49:232–236) on 117 diabetic patients: higher values of urinary NGAL were observed in normoalbuminuric  and increased from micro- to macro-albuminuria, suggesting a tubular damage.

4.2 Biomarkers of therapeutic responce: a very intesting aspect, well described by Rioco-Fontavo et al. (page 6, line253) SGLT2i demostrated, as reported by the authors, “to decrease TNFR-1 and increase branched chain AA, lactate..” but also a significant clinical reduction of albuminuria in DKD patients was also well described.

References: some errors need to be corrected.

At this moment the clinical trials on DKD point attention towards reduction of albuminuria and improvement of eGFR, with the limits already expressed by the authors. However, novel biomarkers are needed in order to identify more accurately patients at high risk of progression to ESKD, taking into accounts that the progression of DKD is usually a slow process that may take decades to clinically emerge.

Author Response

Reviewer 1

  1. References with withdrawn from the abstract.
  2. Phrase suggested by the reviewer added.
  3. The fact that eGFR equations can over or underestimate true GFR was addressed in the manuscript.
  4. We elaborated more on the role of cystatin C in the diagnosis of DKD.
  5. The Carvalho study was mentioned in the paper.
  6. The suggested sentence by the reviewer was added.
  7. The references were fixed.
  8. The manuscript was revised for grammar.

Reviewer 2 Report

This is an interesting review, giving a comprehensive analysis of literature on novel biomarkers of diabetic kidney disease. Discussed evidence is relevant and of quality. Before the manuscript is accepted, I suggest authors clearly define the angle being taken to the reader. For example, within the introduction, introduce similar reviews that have been published on the topic, and clearly state how is the current study different. More emphasizes should also be clearly stated on the flow of the conversation, this will be stating why did you choose to discuss the selected biomarkers. What sources of evidence were accessed for this information. A progressive analysis of literature should be clear between in vivo (preclinical studies) and how they link with clinical data. Importantly, the addition of schematic diagrams can significantly enhance the quality of evidence, these should easily supplement some of the key points/markers discussed within the manuscript.

Author Response

Reviewer 2

  1. We added that our study was unique in the sense that it is a comprehensive review of the inflammatory component of DKD in the introduction, which specific biomarkers mentioned throughout the text.
  2. The paper was proofread and revised for grammar.
  3. Figure 1 was added, which helps with the understanding of the different components of the pathogenesis of DKD.